



# Feedforward-Feedback wake redirection for wind farm control

Steffen Raach[1,3], Bart Doekemeijer[2], Sjoerd Boersma[2], Jan-Willem van Wingerden[2], and Po Wen Cheng[1]

[1]Universität Stuttgart, Stuttgart Wind Energy (SWE), Allmandring 5B, 70569 Stuttgart, Germany

[2]TU Delft, Mekelweg 5, 2628 CD Delft, The Netherlands

[3]sowento GmbH, Donizettistr. 1A, 70195 Stuttgart, Germany

**Correspondence:** Steffen Raach (raach@ifb.uni-stuttgart.de)

**Abstract.** This work presents a combined feedforward-feedback wake redirection framework for wind farm control. The FLORIS wake model, a control-oriented steady-state wake model is used to calculate optimal yaw angles for a given wind farm layout and atmospheric condition. The optimal yaw angles, which maximize the total power output, are applied to the wind farm. Further, the lidar-based closed-loop wake redirection concept is used to realize a local feedback on turbine level. The wake center is estimated from lidar measurements 3 D downwind of the wind turbines. The dynamical feedback controllers support the feedforward controller and reject disturbances and adapt to model uncertainties. Altogether, the total framework is presented and applied to a nine turbine wind farm test case. In a high fidelity simulation study the concept shows promising results and an increase in total energy production compared to the baseline case and the feedforward-only case.

## 1 Introduction

Currently, wind farms are operating with individual optimal turbine settings thus not taking wake interactions into account. This strategy is referred to as greedy wind farm control. The two main wind farm control strategies in which wake interactions are taken into account are axial induction control and wake redirection control (see Boersma et al. (2017) for an overview). In the former, the idea is to deviate the blade pitch angle and tip speed ratio from greedy settings in order to enhance farm performance. Changing these control signals alters, among others, the wind velocity deficit in the turbine's wake hence the power production of downstream turbines. One of the first papers that proposes the idea of axial induction control can be found in Steinbuch et al. (1988). By now, scientific results seem to indicate that by using a currently available steady-state model to evaluate optimal axial induction settings, no power improvement on a farm level can be achieved Annoni et al. (2018). However, recent scientific results indicate that by temporally changing axial induction settings, the farms power output in the therein used wind farm simulators can be improved by using control Ciri et al. (2017); Munters and Meyers (2018). Interestingly, the results in Ciri et al. (2017) seem to indicate that downstream turbine need to deviate from greedy in order to improve the farm's power production while in Munters and Meyers (2018), the control settings of the upstream turbines are temporally changing resulting in an improvement of the farm's power output indicating the necessity for more research. The second wind farm control strategy is wake redirection control and studied in this paper. In this strategy, the goal is to





(partially) curtail the wake around the downstream turbines by using yaw settings such that inflow condition for downstream turbines change and potentially more power can be extracted from the wind. Pioneers work regarding the effect of yawing a turbine on the flow can be found in Clayton and Filby (1982); Grant et al. (1997). Although yaw effects on the flow are still not completely understood, results such as presented in Fleming et al. (2014); Vollmer et al. (2016); Howland et al. (2016) are

providing more insight. To the best of the authors knowledge, Jiménez et al. (2010) presents the first steady-state engineering wake model that describes wake deflection due to yaw. Such types of models have appeared to be useful in wake redirection control Campagnolo et al. (2016); Gebraad et al. (2016); Quick et al. (2017); Fleming et al. (2017b) where the objective is to find the turbine's yaw angles that maximize power production. A tutorial regarding the utilization of steady-state models in control can be found in Doekemeijer et al. (2019). A recent steady-state model has been presented in Bastankhah and Porté-

Agel (2016) whereas this farm model is based on the steady Navier-Stokes equations. This model has been incorporated in the currently online available FLOw Redirection and Induction in Steady-state (FLORIS), an optimization tool that aims to find optimal yaw settings that maximize the farm's power production [1].

The control paradigm when employing previously described steady-state models can be seen as open-loop (feedforward). That is, yaw settings are found using a steady-state model and are then applied to the farm. Assuming that the steady-state

model describes accurately enough the farm's behaviour in yawed conditions and no changes in the atmospheric conditions occur, the open-loop control paradigm could potentially work. However, both assumptions are never completely satisfied. For example, a temporal change in wind direction, which can be seen as a disturbance, will result in non-optimal yaw settings. In this paper we therefore introduce additional local feedback controllers that steer the wakes to set point wake positions. The idea of introducing local yaw controllers to steer the wake to a desired position has been introduced before in for example Raach

et al. (2017c) and successfully tested in a high-fidelity wind farm model Raach et al. (2018). In this work, both feedforward and feedback are combined in order to steer wakes in the farm to desired positions that enhance the power capture of a wind farm.

The paper is structured as follows: First the objectives of the tasks of the presented work are assessed and defined in section 2 and the total control framework is presented. The controllers are describted in section 3: in section 3.1 the feedforward

wake redirection methodology is presented and section 3.2 describes the feedback counterpart that supports the feedforward controller. The effectivity of the combination between feedforward and feedback control is assessed in a simulation study with a nine turbine wind farm layout in section 4. Finally, conclusions are given in section 5.

## 2   Objectives and Control Framework

### 2.1   Objectives

The main objectives of this paper are the following:

– Introduction of the feedforward+feedback wake redirection framework

---

[1]FLORIS is online available at: `https://github.com/TUDelft-DataDrivenControl/FLORISSE_M`





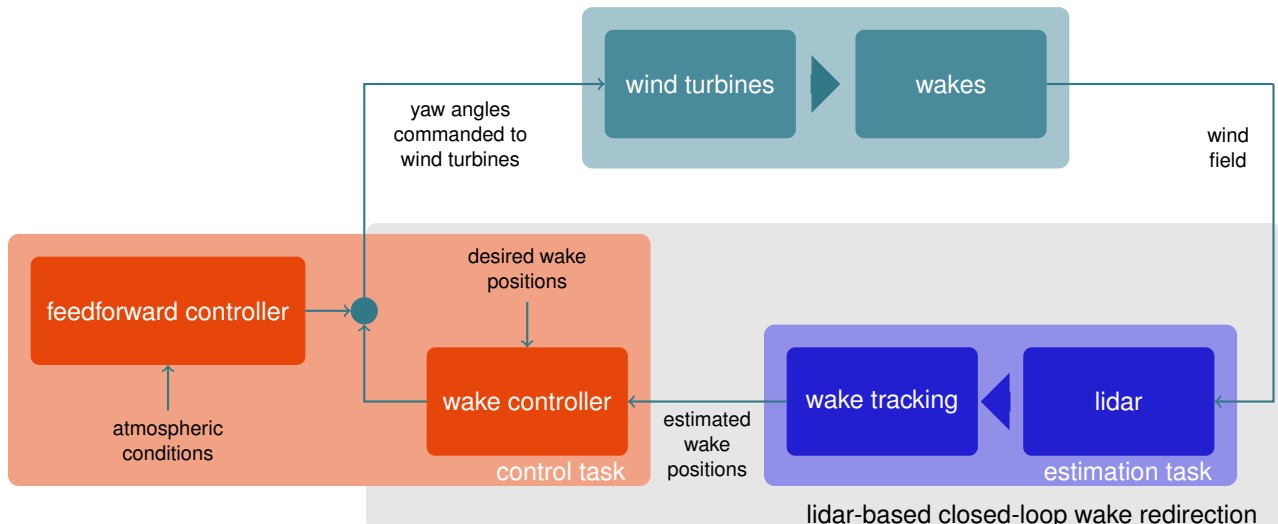

**Figure 1.** The block diagram of the general setup: feedforward-feedback wake redirection control. In the estimation task the lidar system is measuring in the wake of the wind turbine. From the measurement data the wake position is estimated. In the control task the feedforward-feedback controller use the estimated wake position as well as the atmospheric conditions of the setup to provide the yaw angle command to the wind turbine.

– Presentation of the control synthesis of the feedforward+feedback wake redirection

– Demonstration of the concept in an example case in the high-fidelity simulator SOWFA.

## 2.2 Control framework

The framework of feedforward-feedback wake redirection control consists of a feedforward part in which the optimal yaw

5    settings are defined for given atmospheric conditions, e.g. wind speed, wind direction, and turbulence intensity. Further, the feedback controller is continuously active during operation on each wind turbine. Figure 1 depicts a block scheme of the setup. However, in order to use feedback control we need 1) measurements of the farm and 2) a controller design model. The measurements are in this work coming from modeled lidar systems, from which a wake position is estimated. In the employed simulation environment (see section 4.1), perfect lidar systems are modeled that provide estimations of the current

10    wake position. This estimation is used by the feedback controller to reject any disturbance if present. The controller model that is necessary to design the local feedback controllers is estimated from high-fidelity simulation data using system identification techniques.



## 3 Control synthesis

### 3.1 Control task: Feedforward wake redirection control

#### 3.1.1 Methodology

The feedforward controller contains sets of optimized yaw settings. Each set belongs to a specific atmospheric condition and
5 contains yaw angles for each wind turbine in the farm. Each set is evaluated by solving an optimization problem that finds the optimal yaw angles that maximize the power yield of the farm given specific atmospheric conditions. A steady-state (surrogate) wake model is employed in the optimization to obtain the yaw angles.

#### 3.1.2 Surrogate wind farm model

In this work, the state-of-the-art "FLOw Redirection and Induction in Steady-state (FLORIS)" model is used, which is a
10 modular, surrogate wind farm model used in the literature for wind farm control, wind farm topology optimization, and AEP calculations, among others.

   The FLORIS model entails different submodels for single wake deficit, wake summation, and wake deflection due to yaw. In the remainder, the wake deficit and deflection submodels are largely based on the work of Bastankhah and Porté-Agel (2016), and multiple wakes are summed by using the sum-square-of-deficits rule from Katic et al. (1986). Finally, the turbine rotors
are characterized using static mappings for the thrust and power coefficient, usually generated using aero-elastic simulations or more simply based on actuator disk theory. The top-view of the flow field for a 9-turbine wind farm as predicted by FLORIS is shown in Fig. 2. More information on the surrogate model and the general concept of wind farm control using steady-state

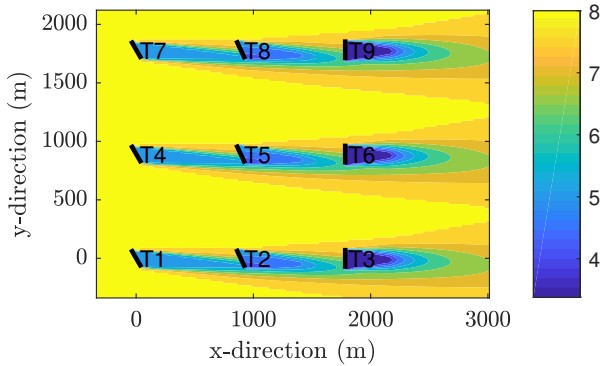

**Figure 2.** The horizontal flow field at the turbine hub height for an example 9-turbine wind farm as predicted by FLORIS

models, including the validation in a high-fidelity simulation environment, can be found in Doekemeijer et al. (2019).





### 3.1.3 Challenges with feedforward wake redirection control

Since the algorithm largely relies on a surrogate model to determine an optimal set of yaw angles, the model's accuracy is very important. Due to the complicated aero- and structural dynamics inside a wind farm, synthesizing such a surrogate model is not straight forward. Assumptions on flow, simplifications in energy extraction, and assuming symmetry in the wake simplify the calculations, however, always introduce uncertainties.

## 3.2 Control task: Feedback wake redirection control

### 3.2.1 Methodology

Closed-loop wake redirection control has been introduced first in Raach et al. (2016) further developed in Raach et al. (2017b). Different control design approaches have been studied in Raach et al. (2017c) and Raach et al. (2017a). So far, the methodology had two main tasks: ensuring the tracking such that the wake is steered to the desired position and adapting to uncertainties and disturbances. In the context of this work, the tasks will be shared, the feedforward controller is responsible to realize the tracking performance and the feedback controller adapts to various uncertainties and does the small adjustments. Therefore, the tuning of the feedback controller differs from previous work because of the different requirements and responsibilities of the controllers.

In this paper, feedback controllers are added to the framework in order to account for undesired wake position deviations. In a perfect setup, wake positions correspond exactly to the optimized yaw settings and hence result in maximum energy yield. These ideal wake positions are referred to as demanded wake positions (see Fig. 1). However, due to for example disturbances, the real wake positions can deviate from the optimal ones and consequently no maximum power production can be ensured. The feedback controller receives wake position deviations (i.e., the difference between demanded and estimated wake position) and evaluates yaw angle deviations accordingly such that the difference between demanded and estimated wake position will be steered to zero. These yaw angle deviations are added to the yaw settings from the feedforward controller, as can be seen in Fig. 1. The design of a feedback controller employs a dynamical model. This model is obtained by running experiments in the simulation model (see section 4.1) and using model identification techniques.

### 3.2.2 Wake position estimation

Since the feedback controller directly relies on a correct estimate of the wake position, the wake position estimation is a crucial part. A downwind facing lidar system is assumed at each wind turbine to measure the wind speed in the wake of the wind turbine. It is assumed that the lidar system can measure the wind speed at a distance of 3 D downstream of the wind turbine. Previous work has shown the feasibility of this assumption, however, challenges remain in realizing it in the field, see Raach et al. (2017b); Fleming et al. (2017a); Annoni et al. (2018).

In this work, a pattern fitting wind field reconstruction methodology is used which assumes a specific shape of the wake. For the wake in a distance of 3 D downwind of a wind turbine it is assumed that the wake can be described as a sum of Gaussian





functions. The basis function

$$\Lambda_j(y_j) = p_{1,j} \exp\left(\frac{-(y_j - p_{2,j})^2}{(2 \cdot p_{3,j}) \cdot^2}\right) \tag{1}$$

is used with $y$ the position vector and the three free parameters $p_{1,j} \ldots p_{3,j}$ which describe the wake deficit, the width of the wake and the $y$ offset of the wake.

For the estimation a sum of several basis functions are combined to

$$\Psi = \sum_j \Lambda_j(y_j), \tag{2}$$

which gives more flexibility in estimating the wake. Furthermore, this assumption also enables to detect overlapping wakes. The resulting wake position is obtained by the weighted mean position of all $N$ considered basis functions

$$y_{\mathrm{res}} = \frac{1}{N} \sum_{j=1}^N \frac{p_{1,j}}{\sum_q p_{1,q}} y_j \tag{3}$$

In the estimation step the lidar measurements are used and fitted to the assumed wake pattern as described in detail in Raach et al. (2017b). Because of the layout a maximum of two overlapping wakes may appear. Furthermore the influence of the wake of a wind turbine at a distance of $10\,\mathrm{D}$ can be neglected. Therefore the choice of two basis functions ($N = 2$) is valid because the wake is measured far downstream and not directly behind the turbine where the shape is not Gaussian. At controller sample time the free parameters are estimated by fitting the measurement data to the wake pattern.

To realize the pattern fitting, the lidar measurement principle needs to be included in the fitting. A lidar remotely measures the wind speed at a defined measurement point by evaluating the back-scattered laser light. However, the measurement principle is a volume averaging around the desired measurement point. The assumption of point measurements is made for describing the measure equation for the pattern fitting. Thus, the lidar measurement at point $[x_i, y_i\, z_i]$ can be written as

$$v_{\mathrm{los},i} = \frac{1}{f_i}\left(x_i u_i + y_i v_i + z_i w_i\right) \tag{4}$$

with the three dimensional flow vector $[u_i, v_i\, w_i]$ at the measurement point and the focal length $f_i = \sqrt{x_i^2 + y_i^2 + z_i^2}$. The wind field model for the wake tracking is defined with the wake pattern of Eq. (2) to

$$\begin{bmatrix} u \\ v \\ w \end{bmatrix}_i = \begin{bmatrix} \Psi \\ 0 \\ 0 \end{bmatrix}. \tag{5}$$





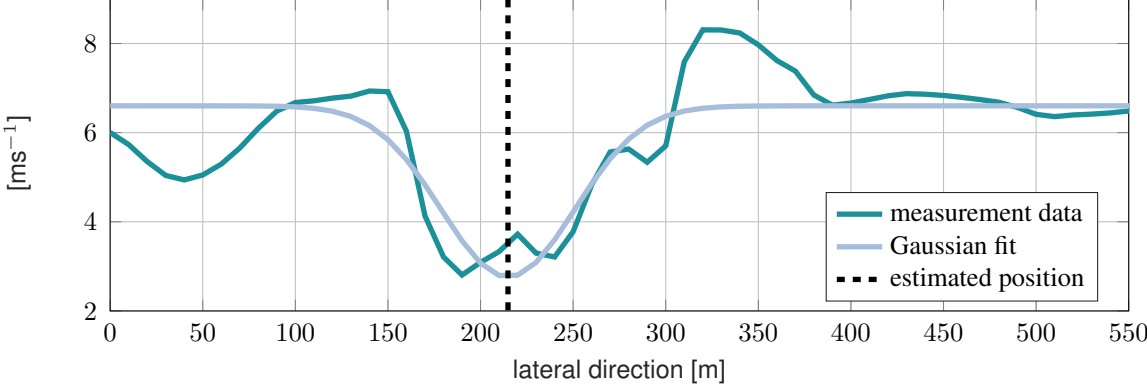

**Figure 3.** An example wake fit to the lidar measurement data using the wake pattern function of Eq. (2) and the wake fitting methods of described in Eq. (6)

.

Hence, the wake tracking is realized by finding the best parameter representation of the following minimization problem

$$\min \begin{bmatrix} (v_{\text{los},1} - v_{\text{los,measured},1})^2 \\ \vdots \\ (v_{\text{los},m} - v_{\text{los,measured},m})^2 \end{bmatrix} \tag{6}$$

with $m$ lidar measurements $v_{\text{los,measured}}$. Figure 3 visualizes the fitting of measurement data at hub height at a downwind distance of $3D$ downwind of the wind turbine. This results in a continuously updated wake position estimation signal which is used in the controller to calculate the desired yaw actions.

The wake tracking method is then applied to all turbines in the case study, however, with constant inflow conditions. In order to get a better understanding of the wakes and the wake tracking a snapshots of all wind turbine wakes and the estimations ar presented. Furthermore, the yaw angles are plotted which are applied in a feedforward appraoch. The results of the wake tracking are later used in the model identification procedure to obtain controller design models. Figure 4 presents the flow measurements at several downwind distances and the obtained wake position estimation result. With the previously described precursor the experiment of an open loop step response is repeated. Lidar wake tracking gives an interesting insight in the wake dynamics and the redirection of it at the turbulent case. Figure 5 presents the result of the wake tracking of the turbulent case of the step response simulation. The signal is also plotted being filtered with a phase free filter to visualize the redirection.

### 3.2.3 Feedback controller design

As previously described, the feedback controller in the combined feedforward-feedback setup mainly has the task to adapt to disturbances and model uncertainties. The feedforward controller is responsible for the main proposition of the yaw command and the feedback controller only rejects disturbances and uncertainties. This approach is also known as the 2 DOF controller

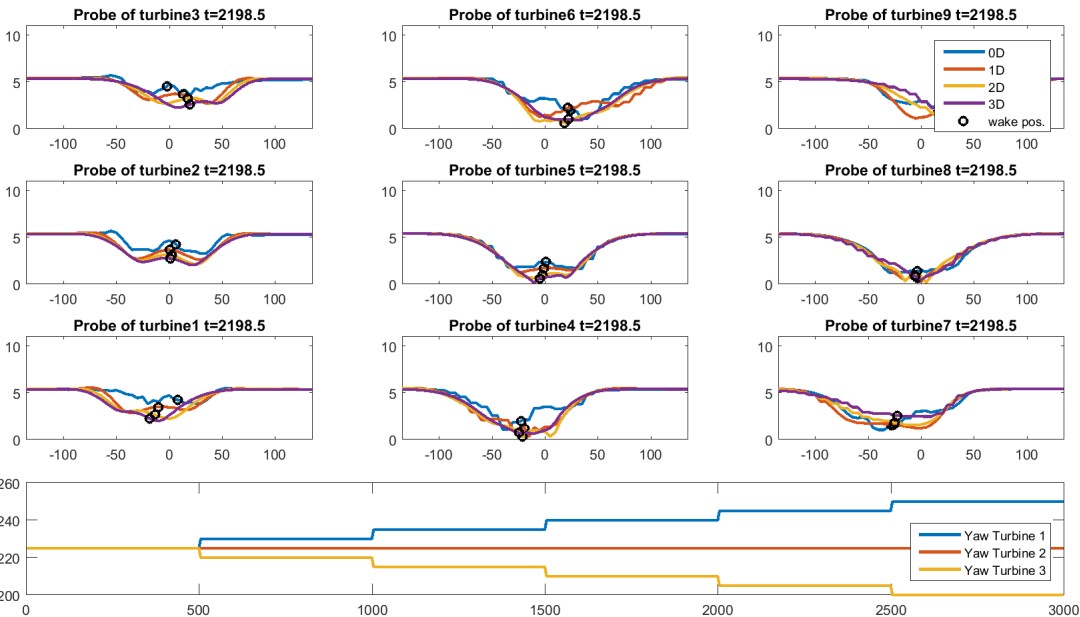

**Figure 4.** Snapshot of an open-loop experiment in SOWFA, where different yaw setpoints are applied to the first row of the $3 \times 3$ wind farm layout. The wake is measured at different downwind distances and estimated using the presented wake tracking approach at each distance.

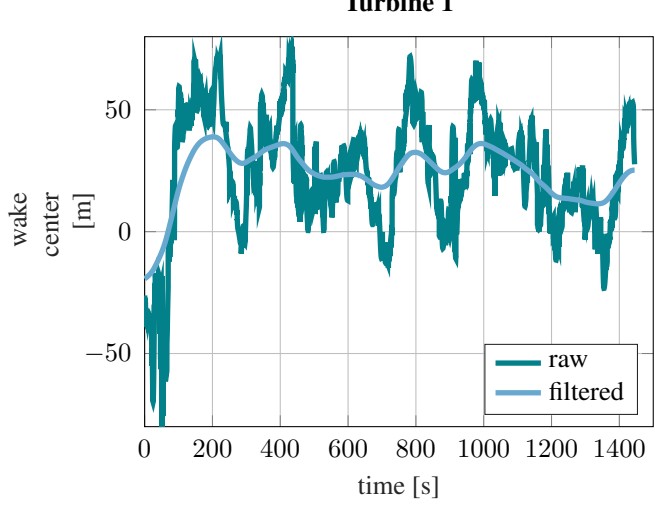

**Figure 5.** Raw time result of the wake tracking of turbine 1 at a position of 2.5 D downstream with the turbulent atmospheric conditions of the example case. To evaluate the wake redirection the result is filtered with a phase free filter. This shows the feasibility to estimate the wake position in more realistic turbulent conditions.



design approach. It allows the the feedback controller to be tuned much softer and robust than in the feedback only case where the feedback is also responsible to guarantee a satisfying tracking performance.

Previous work has investigated different controller synthesizes, like an internal model controller, an $\mathcal{H}_\infty$ controller, and a robust $\mathcal{H}_\infty$ controller. Those controller have resulted in higher order controllers due to the used synthesis methodologies. In this work, the methodology of structured $\mathcal{H}_\infty$ controller design is applied, which is implemented in the Robust Control Toolbox of Matlab, see Apkarian and Noll (2006).

Important performance criteria to design and evaluate the feedback controller are the output sensitivity $\mathcal{S}$, the complementary sensitivity $\mathcal{T}$ and the controller sensitivity $\mathcal{U}$. They quantify the influence of the disturbances or references to the output or the controller. With a given plant model $G$ and the controller $K$, the performance criteria can be evaluated. More precisely, according to Skogestad and Postlethwaite (2005), the sensitivity $\mathcal{S}$ gives the closed-loop transfer function from an output disturbance to the system output, the complementary sensitivity $\mathcal{T}$ is the closed-loop transfer function from the reference to the output and is further the complement of $\mathcal{S}$, and $-\mathcal{U}$ is the transfer function from the disturbance to the control signal. Thus,

$$\mathcal{S} = \frac{1}{1+GK}, \tag{7}$$

$$\mathcal{T} = \frac{GK}{1+GK}, \text{ and} \tag{8}$$

$$\mathcal{U} = \frac{K}{1+GK}. \tag{9}$$

The fixed structure $\mathcal{H}_\infty$ controller synthesis uses the performance weights and a given control structure $K$, e.g. a proportional controller, and solves the mixed sensitivity problem

$$\min_K \kappa$$

$$\text{s.t.} \left\| \begin{matrix} W_\mathcal{S}\mathcal{S} \\ W_\mathcal{T}\mathcal{T} \\ W_\mathcal{U}\mathcal{U} \end{matrix} \right\|_\infty \leq \kappa, \tag{10}$$

where

$$\left\| \begin{matrix} W_\mathcal{S}\mathcal{S} \\ W_\mathcal{T}\mathcal{T} \\ W_\mathcal{U}\mathcal{U} \end{matrix} \right\|_\infty = \left\| \begin{matrix} W_\mathcal{S}(1+GK)^{-1} \\ W_\mathcal{T}GK(1+GK)^{-1} \\ W_\mathcal{U}K(1+GK)^{-1} \end{matrix} \right\|_\infty, \tag{11}$$

with $\kappa$ the bound on the $\mathcal{H}_\infty$ norm and the weights $W_\mathcal{S}(s)$, $W_\mathcal{T}(s)$, and $W_\mathcal{U}(s)$, respectively. for the given control structure $K$ and its free parameter (e.g. the proportional gain). For the closed-loop wake redirection control, a proportional-integral controller structure (PI controller) is used because the integral part adjusts well to model uncertainties and guarantees a zero offset. As mentioned, in contrast to previous feedback-only controllers, for the combined feedback-feedforward approach the



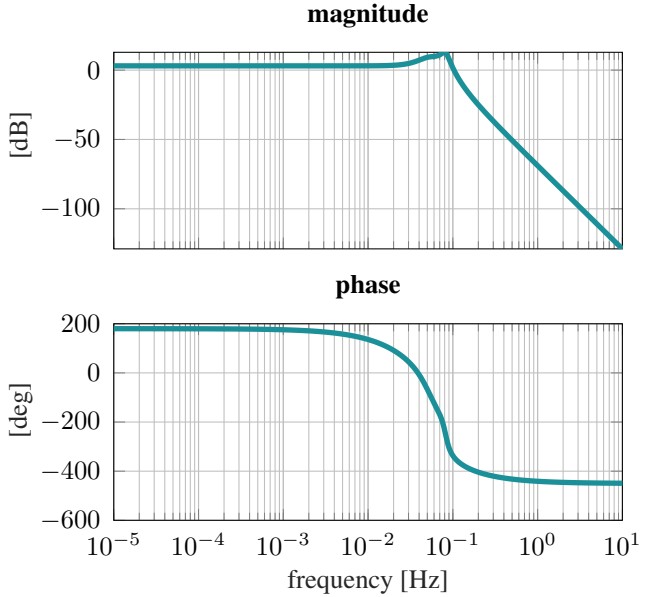

**Figure 6.** The obtained controller design model that was derived from step responses of SOWFA.

feedback controller can be synthesized with a smaller bandwidth. In the following the design process is described. First, a controller design model is derived, then the controller is synthesized using the fixed-structure approach.

The controller design model is derived from open-loop experiments. Different yaw setpoints are set and the flow data is measured with the lidar wake position estimation. Figure 4 shows a snapshot of the wake tracking for the model identification.

5   From theses results a transfer function is estimated that gives the same step response than the estimated wake position from the simulation data. Figure 6 shows the Bode plot of the obtained controller design model.

As a next step, the fixed structure controller synthesis is formed by the performance weights $W_{\mathcal{S}}(s)$, and $W_{\mathcal{U}}(s)$, respectively, as follows:

$$W_{\mathcal{S}}(s) = \frac{s/M + \omega_B}{s + \omega_B A} \tag{12}$$

10  $$W_{\mathcal{U}}(s) = \frac{R^2(s^2 + 1/2\sqrt{2}\omega_d s + \omega_d^2)}{10(s^2 + 25\sqrt{2}\omega_d s + (R\omega_d)^2)} \tag{13}$$

with $w_B = 0.02$, $A = 10^{-7}$, $M = 2$, $\omega_d = 0.005$, and $R = 20$. This setup ensures a good disturbance rejection for low frequencies and no controller action on high frequencies. In the following section, the controller and its resulting sensitivities are analyzed.



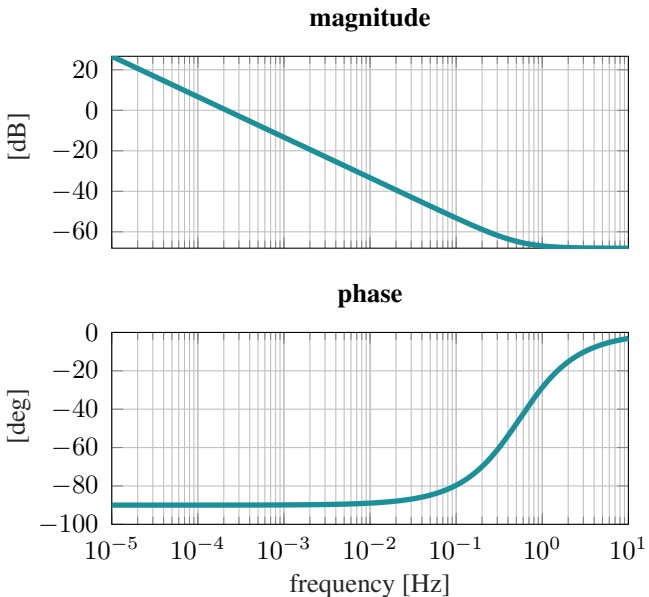

**Figure 7.** The derived fixed structure PI controller is analyzed in the Bode plot.

#### 3.2.4 Controller analysis

The derived controller is shown in the Bode plot in Figure 7. The controller has the following parameter, $K_p = 3.94 \cdot 10^{-4}$, and $K_i = 0.0014$. The performance analysis of the controller is shown in Figure 8, the sensitivity $\mathcal{S}$ and the controller sensitivity $\mathcal{U}$. The sensitivity shows a good damping for low frequencies which results in an offset free control. No static error remains

after any disturbance. The controller sensitivity shows a good roll-off for high frequencies which means that higher frequency movement of the wake is not controlled. Due to the measurement distance downwind of the wind turbine it is needed to adjust the controller in such a way to prevent it from additional control action at higher frequencies, like wake meandering.

### 4 Example case: 3x3 wind farm

#### 4.1 Simulation environment

The high-fidelity wind farm model Simulator fOr Wind Farm Applications (SOWFA), developed by the National Renewable Energy Laboratory, is used to test the proposed control strategy in a wind farm with a regular $3 \times 3$ layout. Figure 9 gives an overview on the spacing and the wind farm layout as well as the inflow wind direction.

SOWFA is based on large-eddy scale simulation techniques and solves the three-dimensional, unsteady, incompressible Navier-Stokes equations over a finite temporal and spatial mesh, accounting for the Coriolis forces. In SOWFA, the larger

scale dynamics are resolved directly while turbulent structures smaller than the spatial discretization are approximated using a subgrid-scale model Churchfield et al. (2012).



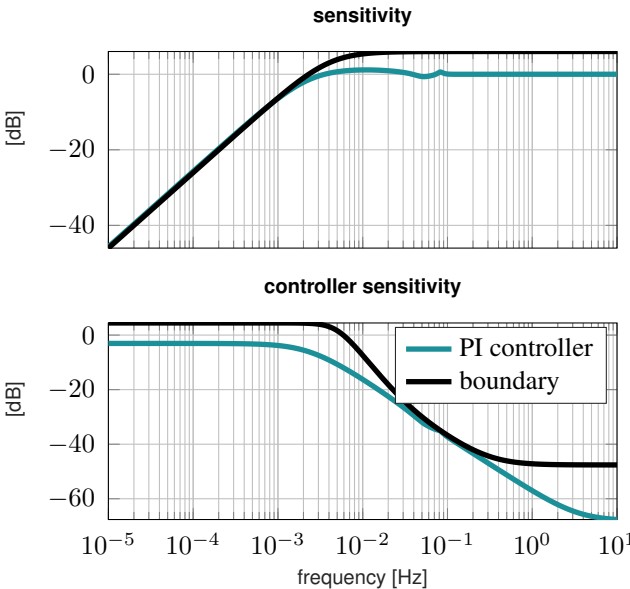

**Figure 8.** Controller analysis: The controller sensitivity and the input sensitivity is analyzed in the two plots. In black, the boundaries of the performance weights are plotted.

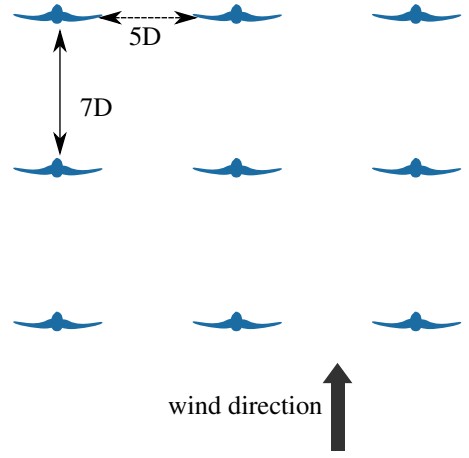

**Figure 9.** The case study wind farm layout, a regular $3 \times 3$ wind farm layout with a downwind spacing of $7\,\mathrm{D}$ and a lateral spacing of $5\,\mathrm{D}$.

The turbine rotor is modeled using an actuator line representation as derived from Sørensen et al. (2002). This model employs a technique in which body forces are distributed along lines representing the blades of the wind turbine. The influence of the rotating blades on the flow field is computed by calculating the local angle of attack and then determining the local forces using tabulated airfoil data. The local forces are then distributed over the blade using a Gaussian filter. Since the ALM is employed, no detailed study regarding fatigue loading can be performed as could be done when employing a turbine model



**Table 1.** SOWFA case study specifications.

| Variable | Value | Variable | Value |
|---|---|---|---|
| Domain size | $5 \times 5 \times 1\,\mathrm{km}$ | Turbine spacing | $5\,\mathrm{D}$ lateral, $7\,\mathrm{D}$ downwind |
| Turbine type | DTU $10\,\mathrm{MW}$ | Cell size outer regions | $10\,\mathrm{m}$ |
| Ambient wind speed | $7.7\,\mathrm{m/s}$ | Atmospheric turbulence | $6\,\%$ |
| Number of turbines | 9 | | |

such as FAST Jonkman and Buhl Jr. (2005). Instead, this work is focused on the control of the flow in a wind farm and thus detailed information on turbines fatigue is not necessary at this stage of the research.

### 4.2 Demonstration study: atmospheric conditions

In order to cover the turbulent flow for the total domain and the inflow of the SOWFA model a proper precursor needs to be run and used. In this work, a precursor, defined and generated in the CL-Windcon project is used, see CL-Windcon (2019).

The accuracy of the results depend on how realistic the inflow and flow-wind turbine interaction are modeled. Therefore, the inflow and initial flow conditions are generated with the same high-fidelity model. To generate a realistic atmospheric condition, the ground surface is set to a specific roughness length. In our case the roughness is set to $z_0 = 0.01\,\mathrm{m}$. The mean wind speed at hub-height is $7.7\,\mathrm{m/s}$ with a turbulence intensity $Ti = 6\,\%$. The precursor domain consist of a box of $l_x \times l_y \times l_z = 5 \times 5 \times 1\,\mathrm{km}$. All lateral boundaries are set to periodic boundary conditions. This means, the outlet flow is recycled to the inflow. For more details, we refer to the deliverable D1.4 of the CL-Windcon project, which is publicly available, see CL-Windcon Deliverable D1.4 (2018). After a simulation time of $1000\,\mathrm{s}$, a cross-wind of $1\,\mathrm{m/s}$ is added at the west boundary to disturb the wake redirection. This modification of the inflow conditions helps to highlight the benefit of the feedback wake redirection controllers.x

The precursor is used as an initial flow field for the 9-turbine case study used throughout this work. Table 1 provides detailed information on the 9-turbine simulation case that is used throughout this work to demonstrate the effectiveness of the proposed control strategy.

#### 4.2.1 Optimal yaw angles for the test scenario

The complete feedforward-feedback solution will be tested on a 9-turbine wind farm in high-fidelity simulation in Section 4. As the mean atmospheric conditions are constant, the feedforward control signals can be calculated a priori. Specifically, the 9-turbine wind farm is depicted in Fig. 2, in which the freestream wind speed is $8.0$ m/s, the freestream turbulence intensity is $6\%$, and the mean wind direction is along the $x$-axis. For simplicity, a brute force approach is leveraged to find the yaw angles





that maximize the power production of the wind farm, leading to

$$\gamma_{\text{FLORIS}} = - \begin{bmatrix} 30° & 25° & 0° \\ 30° & 25° & 0° \\ 30° & 25° & 0° \end{bmatrix}.$$

The yaw angles between the three columns are identical due to symmetry in the wind farm and in the surrogate model. Furthermore, the wake center positions, a quantity directly related to the effectiveness of wake steering, at $3D$ downstream of

each turbine are

$$\Delta y_{\text{wake}} = \begin{bmatrix} -43.2\text{m} & -44.4\text{m} & -11.7\text{m} \\ -43.2\text{m} & -44.4\text{m} & -11.7\text{m} \\ -43.2\text{m} & -44.4\text{m} & -11.7\text{m} \end{bmatrix}.$$

In case of a model mismatch, the predicted wake center locations $\Delta y_{\text{wake}}$ will deviate from the true locations. In the case that the deflection is too small, then more power may be extracted at the downstream turbine by increasing the yaw angle of the upstream turbine. Similarly, if the wake displacement is larger than necessary, then the upstream turbine may capture more

power by decreasing the yaw angle at a negligible loss in power of the downstream turbine.

### 4.3   The simulation cases

In the following three different control cases are compared: a baseline case (BL), a feedforward case (FF), and a feedforward+feedback case (FF+FB).

In the baseline case all wind turbines are aligned with the main wind direction and now yaw actuation is assumed. This case

represents the current operation of a wind farm.

In the feedforward case, optimal yaw angles are computed with FLORIS as described in section 3.1.

In the feedforward+feedback case at each wind turbine the wake position is controlled with the feedback controller and supported by the feedforward controller. A major question in this case is the choice of desired wake position information for each wind turbine. In our case, first, open-loop simulations with the optimized yaw angles using the FLORIS model are

conducted. The wake positions are analyzed and the steady state was then used as commanded information for the feedback controller. In reality a combination of a pre-study with a steady state model like FLORIS or a high fidelity model like SOWFA may give initial values which will be adapted after in a tuning phase with measurement data.

### 4.4   Results

The simulations are initialized with the evolved precursor flow field and the specified inflow conditions. Figure 10 compares

the total power output of the wind farm with the three controller cases. Altogether, the improvements of feedforward and feedforward+feedback are visible and a higher power output is reached compared to the baseline case. Due to the adaption





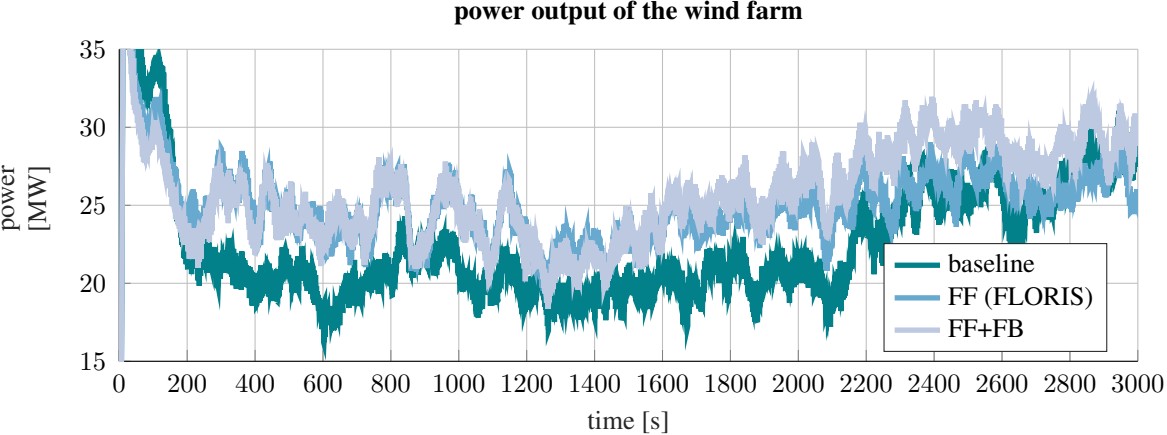

**Figure 10.** Comparison of the total wind farm power output: The baseline case, in which the wind turbines are all aligned with the main wind direction, is compared to the feedforward case, in which the optimized yaw angles of FLORIS are applied, and the feedforward+feedback case, in which additional feedback controllers are controlling the wake position of each wind turbine.

to the changing inflow condition the feedforward+feedback controller over performs the feedforward and baseline cases. The reason for that can be seen in Figure 11 where the yaw angles of turbine 1 and turbine 4 of the different cases are compared. The feedback controllers in case 3 adapt to the changing wind conditions and avoid wind turbine wake interactions. The impact can be seen in Table 2 where the total energy yield with respect to the baseline case is analyzed.

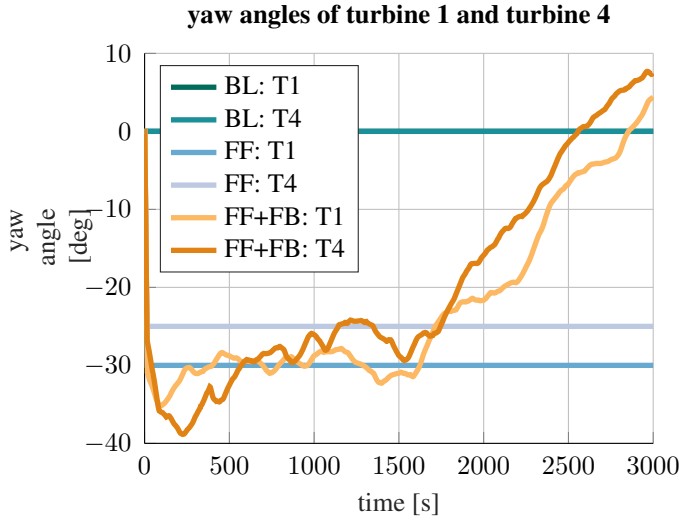

**Figure 11.** Analysis of the yaw angles of wind turbine, which is located in the first row, and wind turbine 4 being in the second row. In the baseline case, the yaw angle is constantly 0 deg, in the feedforward case, a static yaw angle is applied, and in the feedforward+feedback case, the yaw angle is adjusted by the feedback controller.



**Table 2.** A comparison between the total power output of the example case. The table presents the total produced energy of the wind farm with the different control concepts baseline (BL), feedforward (FF), and feedforward+feedback (FF+FB). Furthermore, the power increase with respect to the baseline case is analyzed.

| Case | Total Energy [MWh] | Increase [%] |
|---|---|---|
| BL | 12.4 | - |
| FF | 13.8 | 11.2 |
| FF+FB | 14.5 | 16.9 |

## 5 Conclusions

This paper has presented a combined feedforward-feedback wake redirection control approach. The framework and control approach has been presented in detail. Afterwards, it was adapted to a demonstration case in which a $3 \times 3$ wind farm layout is investigated. Three controller cases are compared to each other: a baseline case, a feedforward case, and a feedforward+feedback case. The feedforward yaw angles are computed using the surrogate model based on FLORIS. For the feedback controller a proportional-integral (PI) controller is investigated and designed using a structured $\mathcal{H}_\infty$ controller synthesis approach.

The control cases are applied to the wind farm using neutral atmospheric conditions and a mean wind speed of 7.7 m/s which is in the partial load region of the wind turbines. Additionally, a cross-wind is imposed to demonstrate the adaptivity of the feedback controller. The combined feedforward+feedback controller adapts to the disturbance. This means the feedback controllers maintain the desired wake and steer the wake to it by adjusting the yaw. Therefore, the enforced wake impingement are mitigated. This results in a higher power output compared to baseline and feedforward only case. Altogether, both controllers, the feedforward and feedforward+feedback, increase the total power yield of the demonstration case compared to the baseline simulation case.

As a next step, changing inflow angles need to be taken into account, as well as changing atmospheric conditions. It further needs to be studied, how these additional changes impact the wake redirection and the feedback controller. Another important question is the choice of wake position set points of the wind turbines in the farm. This point needs further investigation, especially the different calculation methodologies need to be studied.

*Code availability.* The FLORIS model is publicly available at Github[2].

*Competing interests.* There are no competing interests of the authors.

*Acknowledgements.* The CL-Windcon project is acknowledged which has received funding from the European Union's Horizon 2020 research and innovation programme under grant agreement No 727477.

---

[2]https://github.com/TUDelft-DataDrivenControl/FLORISSE_M



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
