# Peer review of "Feedforward-Feedback wake redirection for wind farm control"

_Wind Energy Science, 2019_

## Referee Comment (RC1) · Katherine Dykes (Referee) · 26 Dec 2019

WES 54 Comments: Feedforward-Feedback wake redirection for wind farm control

Overall

- Interesting topic in general and interesting results (not comprehensive enough, but what is provided is interesting).
- The paper needs a lot of work to be honest. See note below on grammar, language style, etc, but there are more substantial issues to address on the overall structure and the content. Detail about suggested areas for restructuring can be found in the section-by-section comments below. For the content, the main concerns have to do with the results section. The approach and methods and results seem to be sound, but it almost feels like unfinished work. Most of the paper is a preamble to what should be a set of rich results and discussion in the Results section. Here it seems like due to lack of time or resources, only an exemplary set of results are shown. Even so, those results are not discussed in enough detail and table 2 which provides numbers and percentages around energy increases makes it seem like the study enables a much more conclusive statement about the performance of the control methods than the actual exemplary results can support. There needs to be one of two things to round out this paper for publication:
    - More simulations run to be able to provide statistics of performance under one (or hopefully more canonical atmospheric conditions)
        - Even for the current exemplary case, you should probably be averaging the power over the period to show the average improved power (and standard deviation) rather than integrating to energy
    - More in depth analysis of the controller performance within the current exemplary case. Go into the data and look at what is happening at different time period and how the additional capability of the FF and FF+FB controllers help improve performance at those different snapshots in time
- It would be good to have an English native edit the document a bit for grammar and language style. There are a large number of grammatical errors and many sentences that are confusing and hard to understand. There are way too many issues to provide detailed feedback throughout the whole document – that is an undue burden to ask of a reviewer. This should have been done before sending the article for review because it honestly is difficult to read at this point.
- Many places in document use vague and non-specific language – always be as precise and quantitative as possible
- Odd that the citations are not buffered by brackets, commas, parentheses or other

Abstract

- Should be two commas in sentence on FLORIS. Second comma after "model"
- More proper to say set of atmospheric conditions
- Sentence on applying yaw angles to wind farm seems incomplete – how are they being applied? Feedforward control?
- What disturbances? What uncertainties? Vague
- A 9 turbine test case – in a simulation environment, what configuration? Grid?

- Promising results? Increase? Again vague

**Introduction**

- Individually optimized turbine settings. Also not self-evident that they don't take wake interactions into account. What do you mean by wake interactions? Be more explicity on greedy wind farm control – where turbines optimize their own performance without regard to the impact to overall wind farm performance
- Farm control can be used to increase power, reduced loads / improve reliability, support the grid or perform other services. Need to be careful to acknowledge that you are focusing on the objective of power performance.
- P. 1. Line 16, steady-state induction not promising for power (unless under very specific conditions – i.e. very closely spaced turbines) but has a lot of promise for achieving other system level objectives (basically all except power). Important to acknowledge that energy production is not everything – the most important but not the only metric. Also, there has been work that has shown that induction control to improve power capture may work in region 2 by adapting the downstream controller in waked conditions (see work from UT Dallas and elsewhere)
- Line 18-22 sound strange – therein used wind farm simulators is odd wording. Also not clear what it means by temporarily changing settings – should be explicit, avoid vague language.
- Line 23, again odd sentence structure and grammatical errors. Suggest: The second wind farm control strategy is wake redirection control and this is the focus of the current work.
  - I can't edit the whole paper at this level of detail. See general note above on having a native English speaker edit the paper for language style and grammar
- P2. Line 1, Not curtailing the week, curtailing means cutting off. It is redirecting the wake. Whole sentence needs rewrite.
- Pioneering work – also what did it do? Vague
- Re: Fleming and other references – what insights? Again vague
- Should have a reference for the FLORIS code in addition to the web link – ask Christopher Bay of NREL for the recommended way to cite
- Not self-evident that FLORIS is feedforward. It can be used in either feedforward or also with feedback. It just so happens that most of the work to date has been feedforward.
- "could potentially work" is vague.
- For example, with respect to the latter criteria on steady atmospheric conditions,"…
- "steer the wakes to set point wake positions" – clarify. It would be nice to see a bit more of an elaboration of the work by Raach in the 2017 and 2018 papers since this paper is building directly from that effort.

**Objectives and Control Framework**

**Objectives**

- This is an odd section to include in a journal article. I would recommend bringing this up into the introduction right before the overview of the structure of the paper. The objective of the work should be very early in the paper (as early as possible without disrupting the flow)

**Control framework**

- I also suggest considering moving this into the introduction section. There are many places in the introduction where you could refer to figure 1 and it would help with clarity. It helps provide specificity and precision to the overall discussion. I also recommend a more detailed discussion of the figure than currently what is in section 2.2 with ore precise language.
- What do you mean by "perfect lidar systems"
- Be more explicit on what disturbances are
- Last sentence is confusing

**Control synthesis**

**Control task on feedforward wake redirection**

**Methodology**

- Again imprecise and vague – what optimized yaw settings, for what conditions? Speed, direction, TI? Lack of detail in introduction on the strategy affects this section.
- Wake redirection is a broad term but then you jump immediately to talking about yaw. If the reader does not know better, this would be interpreted as the same thing
- State that the surrogate model will be described in the next section. If you introduce a term and do not explain it in situ, then you need to say where it is described

**Surrogate wind farm model**

- What do you mean by surrogate? Surrogate has a precise meaning to many people that is usually a purely statistical (or machine-learned) model. FLORIS uses a lot of physics and also has some calibration. It is not a surrogate model but surrogates of FLORIS can be made.
- Provide citation on introduction of FLORIS (same citation as prior software citation that needs to be added)
- P. 4 line 12 contradicts prior statement directly. A surrogate model is one model, not a collection of models. Now you are mixing FLORIS the model with FLORIS the software framework. If you are going to refer to FLORIS as the software framework that collects many models together, make sure you are consistent in that usage versus the older FLORIS model that was the original model as developed by Gebraad and others.
- Line 16, not actuator disk theory, to be more precise you should say "blade element momentum theory". This theory is broadly used to support both aero-elsatic simulation or other more static rotor aerodynamics tools.
- Where does the 9 turbine farm come from? Any references? Who and how was it developed? Why was the spacing and turbine size chose and how?

- Figure 2 caption needs more detailed information and/or more discussion in the text. Its full of content. If you use such a graphic, then make sure to describe it fully.

Challenges with feedforward control

- I'm very confused by this section. It needs a LOT of work. It is again vague and makes strong claims about the FLORIS model(s?) without any concrete evidence. Why / when do the assumptions not hold? The assumptions list seems like a laundry list.

Control task of feedback wake redirection control

Methodology

- "due to, for example," pg. 5 line 17
- Maximum power production can not be ensured. Line 16 use energy yield, line 18 use power, need to make sure not mixing these – important to be clear about which you mean to refer to at that specific instance. Power seems more appropriate when talking about instantaneous / short time durations
- Would be good to clarify and explain WHY you are using the wake center position, and also make sure that you say wake center position when that's what you mean versus wake position. You could have a feedback control system based on other sensing mechanisms, so why lidars and why the wake center?
- Explain dynamical model or say that it will be explained later.

Wake position estimation

- "because of the layout", what layout? The 9 turbine system was not thoroughly introduced, need to elaborate on this statement
- Why can 10D downstream be neglected? Certainly not always the case offshore…
- Statement about model validity with basis of N=2 needs further bolstering and support. Presented arguments are not convincing enough
- Grammar and spelling errors in p. 7 line 7-8
- Figure 4 itself as well as the explanation of it needs a lot of work. Figure 4 are two small so it is very hard to see the plots. Why are you choosing t=2198.5? why do you show the step history of yaw settings if you are only going to show one snapshot in time? Caption not detailed enough.
- "lidar wake tracking gives an interesting insight…" what insight? Vague and figure 5 also could be explained better.

Feedback controller design

- DOF acronym used and not defined on first use
- Much softwer and more robustly
- Syntheses plural
- Prior work mention should include citations, also should explain what the different controller types are, and where does the H-infinity label come from and what does it refer to. And why did you select it over the other approaches

- Is SOWFA defined or explained on first use? Shows up in figure 6 caption.
- It is confusing and unclear what high fidelity simulations are run, what the set up is, what software is being used and how that is then being used for the controller design. Where is this explained?

Controller analysis

- Reported K values without any explanation or interpretation
- Concept of higher frequency content and wake meandering is important but not well developed. What phenomena ocurr at different frequencies in general and which are you trying to control and which are neglected

Example case

Simulation environment

- SOWFA and case are introduced too late. You are referring to both quite a bit in the prior sections so this needs to be somehow up front in the methods. Probably after the FLORIS stuff but before the controller design stuff. Then how and when you are using SOWFA versus FLORIS in your study should also be clarified. The 9 turbine case study needs to be introduced before first use in the FLORIS section figure.

Demonstration study: atmpospheric condition

- Miscellaneous x at the end of p. 13 line 14
- Table 1 introduces a lot of information that should have been used up front. Also, the DTU 10 MW is references in the table without introduction or citation. Where does this case study come from? Need citation
- The introduction of cross-wind at 1000 s is critical and needs to be discussed in much more detail of how and why it is applied

Optimal yaw angles for the test scenario

- Confused here as to which tool you are using to find the optimal. This whole section talks about SOWFA but then the angle matrix has FLORIS as a subscript. Also, if using FLORIS, then why wouldn't you optimize? Why use brute force? If using SOWFA, how do the angles compare to what you would have obtained with FLORIS?
- Symmetry also means that second row of turbines are spaced laterally far enough apart that there isn't an influence of multiple upstream turbines from the first row on a given turbine in the second. If they were spaced more closely would you still expect this symmetry?

Simulation cases

- Elaborate on pg. 14 lines 19-22, especially last sentence

Results

- Adaptation not adaption
- This section needs a lot more development with more rigorous analysis and interpretation of the results
- The energy improvement is for the time period analyzed? That is entirely dependent on the specific inflow conditions and time-history of those conditions used in this study. The results should be averaged over a number of time periods that represent canonical inflow cases… its is weird to provide the results in table 2 without better context. There is absolutely no way to generalize the results based on what is presented, so reporting energy value differences and increases is dangerous.

Conclusions

- P. 16 lines 9-12 the conclusion is providing interpretation that should be provided in the results section
- Future work section needs work – what were the key limitations of the current study and how will you address them in the future? Why couldn't they be dealt with here? Also need to speak to need for field validation of various methods

---

## Referee Comment (RC2) · Anonymous Referee #2 · 26 Jan 2020

**Feedforward-Feedback wake redirection for wind farm control**

The manuscript presents a combined feedforward-feedback wake redirection framework for wind farm control. They used FLORIS to calculate optimal yaw angles for a given wind farm layout and atmospheric condition. They estimated the wake center from lidar measurements 3D downwind of the wind turbines. They also used SOWFA to prove the concept of the work.

In general, the topic is relevant for the wind energy community and the analysis is interesting. However, there are also very significant flaws with the manuscript that both reduce its potential impact and preclude its suitability for publication in WES. Also, the manuscript needs a careful review of the English language. A global re-writing is recommended. There are several places where sentences are not understandable.

Major.

1) Please elaborate on time sensitivity (time resolution).
2) What kind of optimization applied in this study? Is it l-1 or l-2 optimization what is the feasible region for optimization manifolds. What are the constraints of this optimization problem?
3) Equation 6. What is the parameter that minimize the optimization problem?
4) The author applied the proposed control strategy on only one case study (speed 7.7 m/s and Ti =6%). To provide better statistics of performance, the authors need to run many simulations with different parameters.
5) Please provide more details about the simulation. Please comment on the spacing impact on the proposed approach.
6) It is not obvious what the authors implied by the surrogate model.
7) What do you mean by phase free filter? Please add some citations.
8) On page 7, the authors mentioned that "only rejects disturbances and uncertainties", In the current case what the disturbances are?
9) Figure 4, Only one snapshot at t=2198.5 sec is shown, Please show the performance at a different time step.
10) Figure 4 needs axis titles.
11) Please add a range of small frequencies at which control can take action. Please connect them with the Bode diagram.
12) What is the cost function that is used to meet the design specification for a feedforward-feedback wake redirection control approach?
13) This is a mixed sensitivity control approach, Please elaborate on the impact of the time delay on the effective control bandwidth and the attainable lower-band for peaks of S and T.
14) Please identify the number of poles and zeros of the system.
15) What is the limit to push down the peak of S without causing other peaks to occur suddenly and unexpectedly?
16) How the current approach can attenuate the sensor noise for real cases to obtain low gain at high frequencies?

17) Assuming 10D downstream is neglected is little bit rough assumption. Please add a comparison between N=2 and N=3 for verification. You need to consider a large number of wake interactions.

18) Figure 10, the difference in total wind farm power output between FF and FF+FB approaches is only shown at small time window (2100-2800) sec. Before and after this period, the trend is approximately similar. Please discuss this point.

Minor:

There are a large number of grammatical errors and many sentences that are confusing and hard to understand. Please check the manuscript and fix.